# Towards the Identification of Patients’ Needs for Promoting Robotics and Allied Digital Technologies in Rehabilitation: A Systematic Review

**DOI:** 10.3390/healthcare13070828

**Published:** 2025-04-04

**Authors:** Alessio Fasano, Maria Cristina Mauro, Elena Beani, Giovanna Nicora, Marco Germanotta, Francesca Falchini, Arianna Pavan, Valeria Habib, Silvana Quaglini, Giuseppina Sgandurra, Irene Giovanna Aprile

**Affiliations:** 1IRCCS Fondazione Don Carlo Gnocchi, ONLUS, 50143 Florence, Italy; mgermanotta@dongnocchi.it (M.G.); ffalchini@dongnocchi.it (F.F.); apavan@dongnocchi.it (A.P.); vhabib@dongnocchi.it (V.H.); iaprile@dongnocchi.it (I.G.A.); 2Department of Developmental Neuroscience, IRCCS Fondazione Stella Maris, 56128 Pisa, Italy; elena.beani@fsm.unipi.it (E.B.); giuseppina.sgandurra@fsm.unipi.it (G.S.); 3Department of Clinical and Experimental Medicine, University of Pisa, 56126 Pisa, Italy; 4Department of Electrical, Computer and Biomedical Engineering, University of Pavia, 27100 Pavia, Italy; giovanna.nicora@unipv.it (G.N.); silvana.quaglini@unipv.it (S.Q.)

**Keywords:** rehabilitation, patient needs, robotics, ICF, systematic review

## Abstract

**Background/Objectives**: Robotic rehabilitation holds great potential for improving patient outcomes, but the integration of these technologies into clinical practice is hindered by a lack of comprehensive tools for assessing patients’ needs. This systematic review aimed to identify gaps in the current literature, with a focus on methodologies and tools for evaluating such needs, particularly those based on the International Classification of Functioning, Disability, and Health (ICF) framework. **Methods**: A systematic review of qualitative studies published between 2021 and 2023 was conducted, updating a previous (2020) review. Studies were identified through PubMed, Scopus, and Web of Science using inclusion criteria focused on qualitative methods capturing patients’ experiences with robotic and technological rehabilitation devices. **Results**: The review analyzed 19 new studies and 20 from the prior review, revealing a reliance on semi-structured interviews targeting small, heterogeneous populations. No studies employed standardized ICF-based tools, and gaps were noted in the exploration of conditions such as Parkinson’s disease, frailty, or conditions that allowed for multi-device experiences. **Conclusions**: The findings emphasize the necessity for tailored surveys grounded in the ICF framework to completely evaluate the needs of patients suffering from sensory, motor, and/or cognitive disorders caused by different health conditions. This work lays the foundation for more inclusive, effective, and patient-centered robotic rehabilitation strategies.

## 1. Introduction

The integration of technology in the rehabilitation field, such as end-effector robots, electromechanical devices, exoskeletons, wearable sensors, virtual and augmented reality, and others, has gained significant attention in recent years, as it offers the potential to enhance patient outcomes and promote personalized care [1,2,3]. However, the widespread adoption of robotics and other technologies in rehabilitation has been limited so far. Although results, particularly from studies involving neurological patients, are encouraging, such technologies are still only accessible in a limited number of clinical centers, often as part of clinical trials.

To ensure the effective integration of robotics into clinical practice, it is vital to understand the diverse needs of patients who could potentially benefit from rehabilitation technology. Such needs can differ considerably in patients with sensory, motor, and/or cognitive deficits due to different diseases and impairments of varying degrees. Matching these individualized needs with appropriate technology and intervention strategies is crucial in order to optimize clinical outcomes. A comprehensive understanding of patient needs is therefore fundamental for clinicians to determine the best strategies for implementing robotics and technological solutions effectively. Understanding the needs of healthcare practitioners is equally important, as their perceptions, expectations, and readiness to adopt new technologies significantly influence the successful implementation of these innovations in clinical settings. This knowledge not only informs the development of tailored interventions but also ensures that the new technology is truly beneficial and aligns with both patients’ and practitioners’ expectations, thus facilitating broader acceptance, also among healthcare policymakers, and successful integration into clinical practice.

The International Classification of Functioning, Disability, and Health (ICF) framework offers a standardized approach to assess health and functional status [4]. Since its adoption by the World Health Assembly in 2001, the ICF has been utilized across various settings for data collection and health assessments. It serves as a common language and data standard that enhances communication among healthcare providers, researchers, and policymakers [5]. By offering a structured, comprehensive, and versatile framework, the ICF significantly enhances the understanding and measurement of functioning and disability, ultimately improving rehabilitation research and practices. As part of this ongoing development, initiatives like the ICF Core Set Project have emerged [6], aiming to refine the application of the ICF across different health contexts and life stages, further solidifying its role as a vital tool in rehabilitation research and practice.

Moving beyond traditional medical perspectives, the ICF framework shifts attention from the causes of conditions to their impact on daily life, categorizing disability as impairments in body function or structure, activity limitations, and participation restrictions. In particular, it enables a comprehensive description of functional impairments across various domains (such as muscle strength, muscle tone, range of motion of joints, pain, balance), and different activities (such as mobility, writing, communicating, self-care), regardless of the underlying condition [7]. In fact, as highlighted in the literature [8], the acceptability, as well as the recommendation for use, of rehabilitation devices often depend on the patient’s level of disability in specific functional domains and activities. Using the ICF framework, rehabilitation needs can be explored by integrating physical, social, and environmental aspects for a multidimensional assessment of care and interventions.

Rehabilitation tools can then be more accurately tailored to specific patient requirements, leading to better clinical outcomes. ICF-based documentation tools indeed support evidence-based practice by providing a structured approach to assessing patients’ functioning, planning interventions, and evaluating outcomes [9,10]. ICF-based tools may also enable patients to express their rehabilitation priorities more effectively, leading to a greater sense of agency and self-determination [11]. Consequently, they would be better equipped to participate actively in their rehabilitation programs, which is crucial for sustained engagement and success.

Moreover, by offering a unified language and standardized framework, the ICF enhances communication among multidisciplinary teams. This standardization ensures consistency in assessing and documenting patient functioning across different healthcare settings and research studies [12,13].

A previous systematic review by Laparidou et al. (2020) [14] highlighted crucial insights from qualitative studies on end-user experiences with robotic rehabilitation, identifying six main analytical themes: logistical barriers, technological challenges, appeal and engagement, supportive interactions and relationships, benefits for physical, psychological, and social functioning, and expanding and sustaining therapeutic options. Despite the thorough thematic synthesis provided, the authors acknowledged important limitations in the literature, particularly methodological heterogeneity across included studies and limited adoption of standardized frameworks for assessing patient experiences and needs. Laparidou et al. explicitly recommended future research to systematically explore standardized methodologies and assessment tools for capturing user experiences more comprehensively, emphasizing the importance of structured frameworks to ensure consistency and comparability across studies. As Laparidou et al. concluded, participants in the reviewed studies also made recommendations for future use and development of robotic devices and interventions. Systematically identifying and addressing patient and practitioner needs should directly guide the development of tailored, patient-centered rehabilitation technologies, as mentioned above.

The objective of this systematic review was therefore to provide a comprehensive investigation of the current literature on the methodologies used to collect data regarding end-users’ needs in robot- and technology-assisted rehabilitation. Specifically, we aimed to evaluate the different approaches utilized in these studies to understand patients’ needs, and to identify gaps where further development or standardization is needed. To achieve this, we updated the systematic review by Laparidou et al. by extending its scope to include recent findings published between 2021 and 2023, in order to remain aligned with ongoing advancements in the robotic rehabilitation field. For all the above-mentioned reasons, particular emphasis was put on tools based on the ICF, in order to comprehensively capture dimensions related not only to functional impairments (e.g., motor deficits) but also to patients’ participation, perspectives, and daily life implications of their condition.

This work is part of the Italian Initiative “Fit for Medical Robotics” (Fit4MedRob) (https://www.fit4medrob.it/) aimed at implementing robotics and allied digital technologies in clinical practice across different patients’ age groups, from children to the elderly, affected by different diseases. As part of this Initiative, we plan to conduct pragmatic and exploratory trials to evaluate the effectiveness and sustainability of robotics-assisted rehabilitation. In this view, the systematic review presented here serves as a critical step towards the development of surveys designed to capture the needs of patients. These surveys will utilize the insights gathered in this review to more accurately assess and address the rehabilitation requirements of diverse patients’ populations, a starting point in guiding the design of relevant clinical trials on robotic rehabilitation within the scope of the Initiative.

## 2. Materials and Methods

### 2.1. Study Design

This review has been performed following the Preferred Reporting Items for Systematic Reviews and Meta-Analyses (PRISMA) guidelines [15]. The review protocol was registered on PROSPERO (ID: CRD42024617679).

We conducted a systematic review by thoroughly searching relevant literature related to the existing qualitative instruments and tools aimed at collecting the needs of patients, in the context of employing robotic devices within rehabilitation interventions. Within this analysis, we aimed to understand the tools used, the investigated populations (in terms of clinical conditions and countries), the technological devices used by the responders, and the explored themes. Additionally, we aimed to identify any tools specifically based on the ICF.

We identified a work by Laparidou et al. [14], who systematically reviewed studies until August 2020, reporting end-users’ (patients, caregivers, and healthcare professionals) experiences with robotic devices in motor rehabilitation using ad hoc interviews. Since this study focused on a research topic that was coherent with ours, we decided to update Laparidou’s review by extending the literature search time period to between 2021 and 2023. We preferred not to conduct new research, but to update a previous study, since updating systematic reviews is considered, in general, more efficient than starting new systematic reviews when new evidence emerges, as reported in the literature [16].

### 2.2. Selection Criteria

Our review question was: “What are patients’ perceptions of, experiences with, and needs regarding robotic interventions in motor rehabilitation?”. The eligibility criteria encompassed studies focusing on the firsthand experiences and perspectives of patients who underwent motor rehabilitation by integrating robotic interventions. In particular, following the Population, Intervention, Comparison, and Outcome (PICO) framework [17], the population included patients with all kinds of neurological disorders, who have undergone motor rehabilitation that involved a robotic or technology-assisted intervention, without any comparison to other interventions. The main outcome was patients’ perspectives, opinions, and perceptions regarding those robotic interventions in motor rehabilitation.

Only peer-reviewed studies in English, involving humans, were included. Quantitative studies were excluded, as the emphasis lay on qualitative research providing comprehensive narratives of participants’ rehabilitation experiences. From a methodological point of view, the search strategy was performed in the electronic bibliographic databases MEDLINE (PubMed), Scopus, and Web of Science. These databases were searched from the beginning of 2021 to the end of 2023. The query included a combination of two sets of keywords and related terms: (1) robotic and robot-assisted interventions, therapy, and rehabilitation; combined with (2) qualitative research, interview, focus group, experiences, perceptions, attitudes, and views.

Employing an analogous search query of Laparidou et al. [14] to maintain methodological coherence, we sought to ensure that the most recent papers were also included. For the full search strategy used for the Medline database with specific filters, see Table 1. Medical Subject Headings (MeSH) were also employed in building the search query. Table 2 and Table 3 report the search query and filters for Scopus and Web of Science, respectively. Terms were searched in “All Fields”.

### 2.3. Screening Process

All references were independently reviewed and screened by four reviewers. The screening process began with titles and abstracts, which were evaluated for relevance. Final eligibility was determined through full-text screening. For each paper reviewed, the reviewers indicated whether it met the following inclusion criteria: purely qualitative study (e.g., interviews, focus groups, observations); robotic/technological intervention; targeting motor skills/functions; and rehabilitation-focused (excluding activities of daily living, social companions, etc.). Furthermore, studies not involving patients in their investigated sample were excluded in this phase.

Reviewers were required to provide a justification for exclusion. Any disagreements regarding the eligibility of specific references were resolved by involving a fifth member within the review team.

### 2.4. Data Extraction and Synthesis

A structured and pre-tested data extraction form was employed to systematically gather key information from the selected studies. The data extracted from the included studies concerned:source of data—title, authors, year of publication, DOI;sample characteristics—number of participants, diagnosis, target area of rehabilitation;type of robotic device/technology used;method of data collection and analysis employed;use of the ICF framework.

Four independent reviewers extracted data, and a fifth reviewer checked the data extractions for accuracy.

Due to the heterogeneity of the selected studies, we decided not to perform a meta-analysis. Instead, a qualitative analysis was performed, based on the data extracted from the systematic search and aimed at identifying the main tools deployed to catch patients’ perceptions on robotic rehabilitation. Indeed, given that the selected studies were qualitative in nature and primarily focused on interviews with patients or patients and healthcare practitioners, a narrative synthesis approach was adopted to systematically integrate findings across studies. The synthesis process followed the Guidance on the Conduct of Narrative Synthesis in Systematic Reviews [18], ensuring methodological rigor and transparency. Synthesis in tabular form was integrated with the extraction of numerical data where applicable. This approach let us explore and identify the heterogeneity of studies, and compare them on the basis of the above-mentioned key information. As a sensitivity measure, two independent reviewers cross-validated the thematic synthesis. Any discrepancies in theme identification or interpretation were resolved through discussion with a third member. The objective was to evaluate the assessment instruments used in the selected studies and their applications.

## 3. Results

Our search strategy identified a total of 8925 citations (7101 from PubMed, 75 from Scopus, and 1749 from Web of Science). After title screening, 458 papers were included. Subsequent abstract screenings narrowed down the selection to 124 papers for full-text screening. After reviewing the full-text, 105 papers were excluded from the analysis. Reasons for exclusion at this stage were: full-text not accessible (n = 10); only professionals (n = 9); not qualitative study (n = 31); not robotic/technological intervention (n = 14); not motor rehabilitation (n = 8); not rehabilitation focused (n = 30); review study (n = 3). Therefore, 19 studies were included in our final analysis. Figure 1 shows a flowchart illustrating the results of our selection process.

We extended Table 2 featured in Laparidou et al. [14] in a new table (Table 4) summarizing the characteristics of all the reviewed studies, namely, the 20 studies analyzed in Laparidou et al., to which we added the 19 studies included in our investigation. This table collects the aim(s) of the study, the sample (participants involved), the condition and target area of rehabilitation, the robotic devices included, and the method of data collection and analysis employed in each study. Moreover, the use of tools based on the ICF framework was reported. The table presents the key factors that were important for understanding the types of instruments used in the literature to gather the needs of patients (in some studies also the healthcare practitioner’s needs were included).

Our analysis encompassed 39 studies, of which 20 studies [19,20,21,22,23,24,25,26,27,28,29,30,31,32,33,34,35,36,37,38], spanning the period from 2011 to 2020 and reported in Laparidou et al. were included (only those involving patients, or patients and healthcare practitioners, are analyzed here, while those involving only healthcare practitioners were not considered) and 19 studies [39,40,41,42,43,44,45,46,47,48,49,50,51,52,53,54,55,56,57] published between 2021 and 2023 were added.

The studies were conducted in different countries: Canada [22,26,31,32,35,41,44,46,53], USA [25,28,34,38,48,54], the UK [20,27,29,30,36,37,43,50], Turkey [23], Ireland [24,39,40], Germany [19], Switzerland [42,45], the Netherlands [47,55,57], Denmark [49], Spain [51], and Taiwan [52]. One study [33] took place across three countries (Italy, UK, and the Netherlands), whereas another work [21] mentioned a study conducted in three unspecified European Union (EU) countries. 

Overall, these studies included 491patients, in addition to 21 caregivers involved in 4 studies [22,33,35,43] and to 101 operators involved in 10 studies [20,23,27,36,44,45,51,53,56,57], as well as to therapists involved but not counted in 4 studies [21,39,41,42]. For each study, the sample size ranged from 3 to 42 participants and most studies included both men and women.

According to the information provided, there were more male (n = 330) than female (n = 224) participants, while two studies included only male participants [34,38]. Five studies did not report the participants’ gender [20,36,49,53,54]. Participants’ ages ranged from 8 to 88 years.

The majority of studies included patients who had received rehabilitation after stroke (n = 20 studies) [20,21,25,26,29,33,34,36,37,39,40,44,45,47,50,53,54,55,56,57], or brain and/or spinal cord injury (n = 7 studies) [24,28,31,32,38,46,51]. One study included patients with traumatic brachial plexus injuries [48], one study included patients with tetraplegia [49], and one included a frail elderly population [52]. Two studies included children with cerebral palsy [22,35]. Four studies included samples of participants necessitating rehabilitation intervention for different clinical conditions: multiple sclerosis, muscular atrophy, hereditary spastic paraparesis, Bethlem myopathy, cauda equina syndrome [42]; stroke/brain hemorrhage, hemiplegia, other (e.g., accidents, falls, not specified) [10]; stroke, spinal cord injury, hereditary spastic paraparesis [34]; multiple sclerosis, spinal muscular atrophy, muscular dystrophy [41]. The remaining studies included conditions such as physical disability through traumatic injury or illness [23], shoulder instability or rotator cuff-related pain [27], and neuromuscular conditions [30].

Furthermore, different robotic devices were included in these studies: Ekso was used in the majority of studies [24,28,38,40,44,46], ReWalk in three studies [28,31,32], and Lokomat in three studies [22,23,35]. Other types of devices were used such as orthosis in three studies [21,25,48], assistive technologies in six studies [19,39,43,45,52,55], and various other technologies [26,27,29,30,34,37,41,42,47,49,50,51,53,54,56,57]. In studies [20,33,36], the technology was not defined.

To explore users’ experiences and perceptions, most studies utilized individual interviews as their primary method, whilst three studies performed focus groups [26,28,45]. In particular, eighteen studies [22,23,24,25,27,30,31,32,36,40,43,44,46,47,50,51,54,56] conducted semi-structured interviews, [19,39] used structured interviews, and [29,37] used both structured and semi-structured interviews. Two studies used both focus groups and semi-structured interviews [48,57]. Both [38,49] employed semi-structured interviews, but Thomassen et al. [38] employed in-depth interviews guided by thematic areas to explore user experiences of standing and walking with an exoskeleton to capture participants’ lived experience. On the other hand, Kobbelgaard et al. [49] employed iterative interview sessions explicitly focusing on prioritizing activities and contextual factors relevant for exoskeleton use at participants’ homes. One study supplemented semi-structured interviews with the System Usability Scale (SUS) [52], one study with the Quebec User Evaluation of Satisfaction with Assistive Technology (QUEST) [41], while [42] with both SUS and QUEST. One study [21] relied solely on clinical observations for data collection. Another study [34] combined direct observations with semi-structured interviews. Study [33] integrated in-depth interviews with diary entries and photography activities. Lastly, one study [35] employed direct observations alongside semi-structured interviews with parents, as well as interviews and activities involving children.

Only three studies [20,53,55] applied the ICF framework to analyze the outcomes of their semi-structured interviews. In particular, Sivan et al. [20] structured the patient feedback collection process using relevant ICF categories; Forbrigger et al. [53] utilized the ICF to analyze the interaction between patient impairments and environmental factors for home-based device design; and Spits et al. [55] adopted the ICF to comprehensively explore patient experiences across body functions, activities, participation, and contextual domains.

In relation to the analytical and descriptive themes mentioned by Laparidou et al. [14], the articles found from 2021 to 2023 can be included into the six analytical themes that were previously found (logistical barriers; technological challenges; appeal and engagement; supportive interactions and relationships; benefits for physical, psychological, and social functioning; expanding and sustaining therapeutic options). In particular, the primary variables explored in the selected research studies for the period 2021–2023 can be categorized into:usability, safety, accessibility, user acceptance of the technology [40,41,42,45,47,49,50,51,52,54,57];user perceptions of effectiveness, personal experiences and satisfaction while using the technology [39,40,41,42,44,45,46,53,54,55,56,57];need for professional supervision [50];user requirements for the design of the technology [48,49,51];adaptability of the technology to home and community settings [42,49,50,52,53,57];social and cultural impacts of the technology, i.e., impacts on function, independence and dignity of users [43,46].

**Table 4 healthcare-13-00828-t004:** Study characteristics of all reviewed studies, including those reported and analyzed in Laparidou et al. [14] until 2020, as well as the 19 studies selected in our investigation from 2021 to 2023 (light grey shade). The column “Study” reports the paper under study and the country in which the investigation was conducted. ICF: International Classification of Functioning, Disability, and Health.

Study	Aim(s)	Sample	Condition and Target Area of Rehabilitation	Robotic Device	Method of Data Collection and Analysis	Use of the ICF Framework
Eicher et al., 2019 [19]; Germany	To identify differences regarding usability, acceptability, and barriers of usage of a robot- supported gait rehabilitation system between a younger and older group of patients with gait impairments	13 patients	Stroke/brain hemorrhage, hemiplegia, other (e.g., accidents, falls, not specified); gait rehabilitation	Hybrid Assistive Limb exoskeleton	Structured interviews; qualitative content analysis by Mayring (2010) [58]	no
Sivan et al., 2016 [20]; UK	To evaluate the ICF as a framework to ensure that key aspects of user feedback are identified in the design and testing stages of development of a home-based upper limb rehabilitation system	17 patients and 7 physiotherapists and occupational therapists	Stroke; upper limb rehabilitation	Not defined	Face-to-face semi-structured interviews; analysis based on the updated International Classification of Functioning, Disability and Health (ICF) linking rules and core set categories	only for the analysis
Ates et al., 2014 [21]; Three EU countries (unspecified)	To report on the technical challenges presented by the use of SPO and the feedback from therapists and patients	24 patients; no information about the therapists	Stroke; hand impairment	SCRIPT Passive Orthosis	Clinical observation and descriptive summary into themes	no
Beveridge et al., 2015 [22]; Canada	To explore the experiences and perspectives of parents whose young, ambulatory children with CP were undergoing Lokomat gait training, and consider how parents’ values about walking influenced therapy decisions for their children	5 mothers and 1 father of 5 children	Cerebral palsy; walking rehabilitation	Lokomat	Individual, semi-structured, face-to-face interviews; followed the Dierckx de Casterle approach to analysis of qualitative data: the Qualitative Analysis Guide of Leuven (QUAGOL)	no
Bezmez and Yardimci, 2016 [23]; Turkey	To explore the role of a robotic gait training device and its role in rehabilitation in Turkey	42 participants (7 doctors, 2 nurses, 2 physiotherapists, 2 non-medical personnel, 20 in-patients, and 9 former patients)	Traumatic injury or illnesses; bodily disability and inability to walk	Lokomat	Individual, semi-structured interviews; no information provided on the method of analysis	no
Cahill et al., 2018 [24]; Ireland	To gain an understanding of the experience of using a RWD within a gym-based setting from the perspective of non-ambulatory individuals with SCI	5 patients	Spinal cord injury; walking rehabilitation	Ekso™	In-depth semi-structured interviews; thematic analysis	no
Danzl et al., 2013 [25]; USA	To investigate the feasibility of combining tDCS into the LE motor cortex with novel locomotor training to facilitate gait in subjects with chronic stroke and low ambulatory status; to obtain insight from participants and their caregivers to inform future trial design	8 patients	Stroke; lower limb (gait) rehabilitation	Robotic gait orthosis	Semi-structured interviews; inductive thematic analysis	no
Elnady et al., 2018 [26]; Canada	To describe users’ perceptions about existing wearable robotic devices for the upper extremity; identify if there is a need to develop new devices for the upper extremity and the desired features; and to explore obstacles that would influence the utilization of these new devices	8 people with stroke; 8 therapists: 4 physiotherapists, 2 occupational therapists; 2 rehabilitation assistants	Stroke; upper limb rehabilitation	Wearable Robotic Devices for the upper extremity	Focus groups; thematic analysis	no
Gilbert et al., 2018 [27]; UK	To determine whether or not the MUJO System was acceptable to patients with shoulder dysfunction and their rehabilitation professionals	10 patients and 7 physiotherapists	Shoulder instability (n = 6) and rotator cuff-related pain (n = 4); rehabilitation of the rotator cuff muscles (bi-articular muscles or multiple axial joints)	MUJO System	Semi-structured interviews; Directed Content Analysis was undertaken to organize the qualitative data according to the four constructs of Normalisation Process Theory (NPT)	no
Heinemann et al., 2020 [28]; USA	To describe appraisals of robotic exoskeletons for locomotion by potential users with spinal cord injuries, their perceptions of device benefits and limitations, and recommendations for manufacturers and therapists regarding device use	35 patients	Spinal cord injuries; gait rehabilitation	Robotic exoskeletons (Ekso Indego, ReWalk)	Focus groups; thematic analysis	no
Hughes et al., 2011 [29]; UK	To understand the stroke participants’ experiences of using the novel combination of a robotic arm and iterative learning control system, and to gain greater insight into how systems might be improved in the future	5 patients	Stroke; upper limb rehabilitation	Robotic workstation	Structured and semi-structured interview. Two types of data were collected: comments were recorded during the time when participants were receiving the intervention and immediately following the clinical study, an interview-based question set was used; content analysis	no
Kumar and Phillips, 2013 [30]; UK	To explore the views, experiences, benefits, and difficulties that users of one specific type of PMAS perceive, and to determine which areas of daily life they are used in	13 patients	Neuromuscular conditions; upper limb rehabilitation	Powered mobile arm supports	Semi-structured interviews; thematic analysis	no
Lajeunesse et al., 2018 [31]; Canada	To present the perspectives of individuals with ASIA C or D incomplete SCI concerning the usability of lower limb exoskeletons to R&D engineers and clinicians working in motor rehabilitation	13 patients	Incomplete spinal cord injury; lower limb rehabilitation	ReWalk exoskeleton	Individual, semi-structured interviews; inductive thematic analysis	no
Manns et al., 2019 [32]; Canada	To explore the expectations and experiences of persons with spinal cord injury through training with the ReWalk exoskeleton	11 patients	Traumatic spinal cord injury; standing and walking training	ReWalk exoskeleton	Semi-structured interviews; thematic analysis	no
Nasr et al., 2015 [33]; UK, Italy and Netherlands	To examine stroke survivors’ experiences of living with stroke and with technology in order to provide technology developers with insight into values, thoughts, and feelings of the potential users of a to-be-designed robotic technology for home-based rehabilitation of the hand and wrist	10 patients and 8 caregivers	Stroke; upper limb rehabilitation	Not defined	Application of qualitative methods such as in-depth interviews as well as using diaries and photography activities; thematic analysis	no
O’ Brien Cherry et al., 2017 [34]; USA	To determine participants’ general impressions about the benefits and barriers of using RT devices for in-home rehabilitation	10 veterans	Stroke; upper or lower limb impairments	Hand Mentor™ and Foot Mentor™ devices	Direct observations and semi-structured interviews; inductive thematic analysis	no
Phelan et al., 2015 [35]; Canada	To investigate the expectations and experiences of children with CP in relation to robotic gait training using the Lokomat Pro	5 children and their parents (3 mothers and 2 fathers)	Cerebral palsy; gait rehabilitation	Lokomat Pro	Observations during sessions, semi-structured interviews with parents and use of a customizable “toolbox” of age-appropriate child-friendly techniques; thematic analysis	no
Sweeney et al., 2020 [36]; UK	To understand user perceptions in order to explain low uptake of upper limb rehabilitation interventions after stroke in clinical practice within the National Health Service (NHS Scotland)	8 patients	Stroke; upper limb rehabilitation	Not defined	Semi-structured interviews; thematic analysis	no
Tedesco Triccas et al., 2018 [37]; UK	To explore views and experiences of people with subacute and chronic stroke that had previously taken part in a randomized controlled trial involving tDCS and RT for their impaired upper limb	21 patients	Stroke; upper limb rehabilitation	Armeo Spring	Structured and semi-structured interviews involving open questions; thematic analysis	no
Thomassen et al., 2019 [38]; USA	To generate new knowledge regarding user experiences of standing and walking with Ekso	3 patients	Spinal cord injury (due to traumatic and non-traumatic reasons); standing and walking training	Ekso™	In-depth interviews in a phenomenological tradition; systematic inductive content analyses	no
Shore et al., 2022 [39]; Ireland	To explore insights expressed by a cohort of older adults related to their life experience, their experiences using or assisting someone with assistive devices, and their perceptions of robots and robotic assistive devices	24 patients; no information about the therapists	Stroke; lower limb rehabilitation	Assistive devices	Structured interviews	no
McDonald et al., 2022 [40]; Ireland	To explore the usability and acceptance of RAGT in an acute hospital setting, and to examine users’ perceptions of two different modes of robotic assistance provided during rehabilitation	10 patients; no information about the therapists	Stroke; gait training	Ekso™	Semi-structured interviews of end-user perspectives and two 10-point Likert scales rating	no
Lebrasseur B.Erg et al., 2021 [41]; Canada	To evaluate the usability of actuated arm support	9 patients; no information about the therapists	Multiple sclerosis, spinal muscular atrophy, muscular dystrophy; upper limb rehabilitation	Gowing power-assisted arm support	Quebec User Evaluation of Satisfaction with assistive Technology (QUEST) and semi-structured interviews	no
Basla et al., 2022 [42]; Switzerland	To investigate end-user perspectives and the adoption of an exosuit in domestic and community settings	7 patients; no information about the therapists	Multiple sclerosis, spinal muscle atrophy, spastic paresis, Bethlem myopathy, cauda equina syndrome; walking rehabilitation	Myosuit	SUS and QUEST and one personalized questionnaire, semi-structured interview	no
Hampshire et al., 2022 [43]; UK	To gather users’ and caregivers’ perspectives on the assistive device	6 patients and 2 caregivers	Stroke, spinal cord injury, hereditary spastic paraparesis; walking rehabilitation	Assistive devices	Semi-structured interview	no
Louie et al., 2022 [44]; Canada	To explore the experience and acceptability of an exoskeleton-based physiotherapy program for non-ambulatory patients from the perspective of patients and therapists	14 patients; 6 physiotherapists	Stroke; gait training	Ekso™	Semi-structured interviews and thematic analysis	no
Bauer et al., 2021 [45]; Switzerland	To assess the usability of a prototype assistive therapy chair (T-Chair) that induces exercise stimuli to improve trunk control and standing and walking early after stroke in a rehabilitation setting	15 patients; 11 physiotherapists	Stroke	T-chair device	Focus groups; customized questionnaire	no
Charbonneau et al., 2021 [46]; Canada	To increase our understanding of SCI patient experience using a robotic exoskeleton in the acute post-injury period	9 patients	Spinal cord injury	Ekso™	Semi-structured interviews	no
Nieboer et al., 2021 [47]; Netherlands	To assess attitudes towards “Stappy” in people after stroke to practice walking performance independently at home	17 patients	Stroke	Stappy (sensor-feedback device)	Semi-structured interviews	no
Webber et al., 2022 [48]; USA	To explore patient perspectives on the use of a Mioelectric elbow orthosis (MEO) following surgical treatment of a traumatic brachial plexus injuries (BPI)	8 patients	Traumatic brachial plexus injuries	Mioelectric elbow orthosis	Focus groups; semi-structured interviews	no
Kobbelgaard et al., 2021 [49]; Denmark	Identify users’ needs and preferences for the design of an arm exoskeleton	9 patients	Tetraplegia	EXOTIC arm exoskeleton	Semi-structured interviews	no
Kerr et al., 2023 [50]; UK	To assess the feasibility of a technology-enriched rehabilitation gym (TERG) approach and gather preliminary evidence of its effect on future research	26 patients	Stroke	TERG (e.g., virtual reality treadmills, power-assisted equipment, balance trainers, and upper limb training systems)	Semi-structured interviews	no
Herrera-Valenzuela et al., 2023 [51]; Spain	To determine a comprehensive set of requirements, perceptions, and expectations that people with spinal cord injury (SCI) and the clinicians in charge of their rehabilitation have regarding the use of wearable robots (WR) for gait rehabilitation	15 patients; 10 clinicians (5 physical medicine and rehabilitation physicians and 5 physiotherapists)	Spinal cord injury	Lower limb wearable exoskeletons	Semi-structured interviews	no
Chang et al., 2023 [52]; Taiwan	To develop a novel smart somatosensory wearable assistive device (called SSWAD), combined with sEMG and exergame software and hardware technology for rehabilitating the lower limb muscles of the older adult	25 patients	Elderly frail	Smart Somatosensory Wearable Assistive Device (SSWAD)	Semi-structured interview and SUS questionnaire	no
Forbrigger et al., 2023 [53]; Canada	To investigate the needs of stroke survivors and therapists, and how they may contrast, for the design of robots for at-home post stroke rehabilitation therapy	10 patients; 6 therapists (5 physiotherapists and 1 occupation therapist)	Stroke	Upper limb robotic devices (FitMi, MusicGlove)	Semi-structured interview	only for the analysis
Bhattacharjya et al., 2023 [54]; USA	To evaluate the participants’ approach and nature of engagement with the mRehab system and corresponding changes in outcome measures, and to identify the self-reported factors that influenced the use of the mRehab system at home by community-dwelling individuals with chronic stroke	6 patients	Stroke	mobile Rehab	Semi-structured interview	no
Spits et al., 2022 [55]; Netherland	To investigate stroke survivors’ experiences regarding training using the hoMEcare aRm rehabiLItatioN (MERLIN) system, an assistive device and telecare platform	11 patients	Stroke	MERLIN system	Semi-structured interview	only for the analysis
Lee et al., 2022 [56]; not specified	To present detailed design specifications and exploratory evaluations of an AI and robotic coach capable of monitoring and guiding post-stroke survivors in self-paced physical rehabilitation therapy	5 patients; 4 therapists (3 occupational therapists, 1 physiotherapist)	Stroke	Robotic coaches	Semi-structured interview	no
Langerak et al., 2023 [57]; Netherlands	To define the user requirements for home-based upper extremity rehabilitation using wearable motion sensors for subacute stroke patients	17 patients; 21 therapists	Stroke	Wearable motion sensors	Focus groups and semi-structured interview	no

## 4. Discussion

In this study, we systematically reviewed the existing literature to evaluate methodologies employed for understanding patients’ and healthcare practitioners’ needs in the robotic rehabilitation field. Our investigation focused on qualitative tools and instruments, with particular attention paid to the application of the International Classification of Functioning, Disability, and Health (ICF) framework. The ICF’s classification system encompasses various components, including body functions and structures, activities and participation, and personal and environmental factors. This framework provides a comprehensive assessment that is particularly relevant for rehabilitation, where patient-centered care requires a holistic evaluation of impairment levels and functional capabilities across different domains.

This systematic review identified 19 new studies from 2021 to 2023, in addition to the 20 previously reviewed by Laparidou et al. (2020) [14], thus providing an updated perspective on current trends and limitations in this research area. Building upon the previous review, we also investigated whether qualitative studies had structured their findings using the ICF framework, as recommended for capturing a holistic view of disability, daily life activities, and participation.

As shown, the literature employs mainly semi-structured interviews targeting small populations often with different pathologies and referring to the use of a single device. In particular, no studies used questionnaires/tools based on the ICF. Three studies only [20,53,55] applied ICF to analyze the outcomes of their assessments, and in [20], the authors asserted that the categories of the ICF Comprehensive Core Set could serve as a foundation for structuring interviews to capture user feedback. Nonetheless, as far as we are aware, they have not made such a tool accessible. The possibility of investigating the degree of impairment of different functions through ICF is an extremely important topic, but the use of the ICF in robotic rehabilitation remains unexplored. Indeed, as can be seen in the literature, the acceptability and usability of devices are strongly influenced by the severity and type of disability [43]. Given that functional impairments vary significantly across different domains, rehabilitation technologies should be tailored accordingly. The ICF provides a structured method to assess these aspects: the level of disability can be referred to individual functional domains—such as gait, balance, upper limb function, cognitive status, and speech—for patients with different diseases. A multidomain assessment would allow for a more individualized and precise adaptation of rehabilitation technologies. For instance, stroke patients with predominant gait impairments may require different technological interventions compared to those with cognitive deficits. Furthermore, stroke patients rarely experience impairments confined to a single domain, but they often present with a combination of motor, sensory, and cognitive deficits that affect their mobility, balance, coordination, executive functioning, and even language abilities. The ICF framework allows for classification across diverse health conditions while considering all these factors, as well as environmental and personal influences, which are crucial for designing patient-specific interventions. In fact, rehabilitation should not only be designed to restore movement but also to enhance functional independence and social participation. By failing to incorporate this framework, current research may lack the depth and consistency necessary for developing effective rehabilitation technologies that address, and are tailored to, the full spectrum of patient needs.

In addition, various studies in the literature employed the same survey to examine the requirements of both patients and practitioners. This raises concerns about the adequacy of these approaches in capturing the unique perspectives of each group. In our opinion, to comprehensively capture a broad sample and acknowledge the distinct perspectives and expertise of patients and practitioners regarding technologies, distinct and specific tools are desirable. Indeed, while some aspects of usability and accessibility may overlap, patients and healthcare professionals often have distinct priorities when evaluating rehabilitation technologies. Patients may focus more on usability, comfort, and impact on daily life, whereas healthcare professionals prioritize clinical effectiveness, ease of integration into therapy routines, and cost-effectiveness.

Our review also highlighted a series of limitations in the current knowledge that could be obtained regarding patient needs in relation to robotic and technological rehabilitation. First, no studies have investigated the needs of patients and caregivers using large-scale cohorts. The sample sizes across the included studies ranged from 3 to 42 participants, limiting the generalizability of findings. Second, although studies predominantly focused on major neurological conditions—such as stroke, spinal cord injury, multiple sclerosis, traumatic brain injury, cerebral palsy, and neuromuscular disorders—there is a notable lack of studies focused on the needs of individuals with Parkinson’s disease, limb amputations, cancer, and frailty. Third, the devices used by the study populations are, when specified, only robotic devices for upper limbs and gait training. No studies explored the experience of patients and therapists who used devices for balance treatment or devices for cognitive function treatment. Moreover, no studies have explored the experience of subjects who had used multiple devices simultaneously. Finally, a further research gap concerns the geographical distribution of studies. Few studies have investigated the specific needs of Italian patients and practitioners. The healthcare system in Italy presents unique challenges and opportunities linked to regulations and legal frameworks, such as variations in regional policies, duration of hospital stays, and access to home-based rehabilitation services. The healthcare needs of patients in Italy may therefore vary from those of populations in other countries. Understanding these contextual factors is crucial for designing technologies that align with local healthcare structures and patient expectations.

To address these gaps, the current systematic review represents a foundational step in the development of tailored surveys aimed at capturing patient needs more comprehensively. These surveys should leverage the insights gained from our analysis, particularly regarding the applicability of the ICF framework, to provide a standardized and holistic approach to assessing rehabilitation requirements. Importantly, these surveys may form the basis for pragmatic and exploratory trials under the Fit4MedRob Initiative, which in turn will gather critical data to inform future rehabilitation research and robotic technology development. By aligning these efforts with the Initiative’s objectives, it would be possible to foster a deeper understanding of patient and practitioner needs, ultimately enhancing the integration and effectiveness of robotics in rehabilitation practice.

Despite following a rigorous methodology, this review presents several limitations that should be acknowledged. The qualitative nature of the studies reviewed makes the findings more interpretative rather than statistically generalizable. Indeed, this review did not incorporate a structured risk of bias assessment. Due to the small number of included studies and their heterogeneity in study design, population, and rehabilitation technology, we did not perform a sensitivity analysis to evaluate how excluding specific studies might affect the overall findings.

## 5. Conclusions

This systematic review examined the methodologies used to assess users’ needs in robotic rehabilitation and identified key gaps in standardization, such as the lack of structured assessment tools. While robotic technologies offer significant potential, the absence of systematic, validated methodologies limits our ability to comprehensively capture patients’ and therapists’ perspectives. To address these gaps, we propose developing tailored surveys grounded in the ICF framework, designed separately for patients and therapists to ensure that their distinct perspectives are accurately captured. These surveys will inform future research, enabling clinical trials better aligned to patients’ real-world needs, and eventually fostering the integration of robotics into clinical practice.

## Figures and Tables

**Figure 1 healthcare-13-00828-f001:**
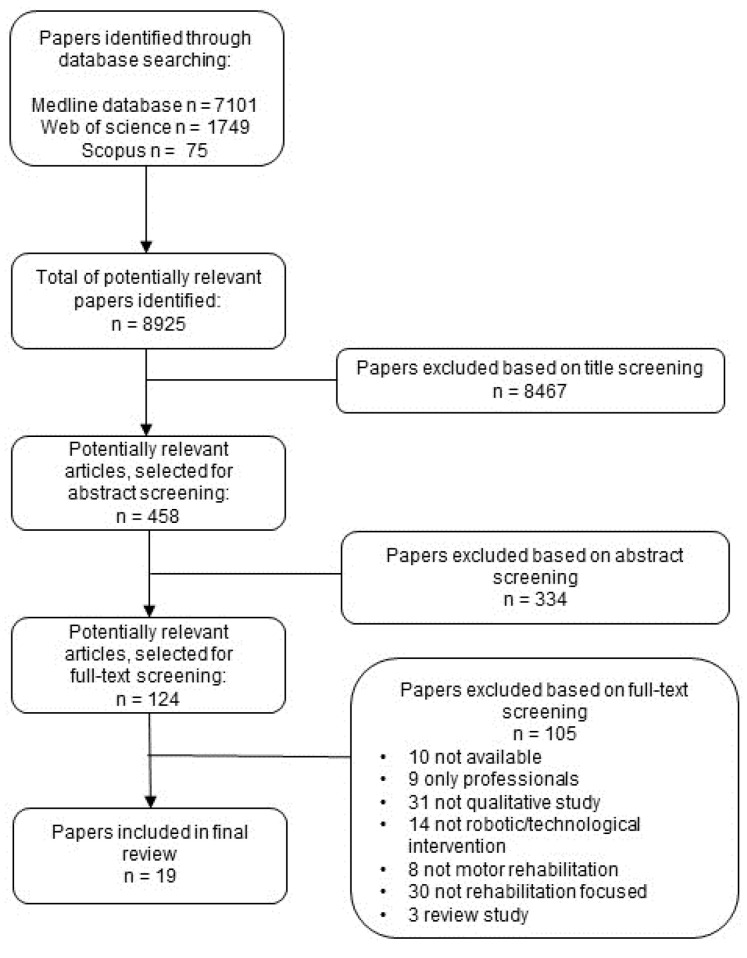
Flowchart illustrating the results of our selection process.

**Table 1 healthcare-13-00828-t001:** Our search strategy in PubMed and applied filters.

**S1**	robotic* OR robot* OR robotic therap* OR robot-assisted OR robot assisted OR exoskeleton* OR assistive robotic* OR walking robotic device* OR personal care robot* OR medical robot* OR assistive OR assistive automation OR wearable robot* OR orthotic* OR orthosis OR exoskeletal* OR exo OR end-effector OR haptic* OR robot regulation*	
**S2**	rehab* OR intervention* OR treatment* OR therap* OR program* OR strateg* OR training OR physiotherap* OR physio-therap* OR “physiotherap*” OR “physical therap*”	
**S3**	Qualitative research OR qualitative OR interview* OR focus group* OR ethno* OR phenomenolog* OR hermeneutic* OR grounded theory OR narrative analysis OR thematic analysis OR lived experience* OR life experience*	
**S4**	(MH “Qualitative Research”) OR “Qualitative research”	
**S5**	S3 OR S4	
**Searched**	S1 AND S2 AND S5	
**Applied Filters**		*Published*: 2021–2023
		*Language*: English
		*Species*: Humans

**Table 2 healthcare-13-00828-t002:** Our search strategy in Scopus and applied filters.

**S1**	robotic* OR robot* OR robotic therap* OR robot-assisted OR robot assisted OR exoskeleton* OR assistive robotic* OR walking robotic device* OR personal care robot* OR medical robot* OR assistive OR assistive automation OR wearable robot* OR orthotic* OR orthosis OR exoskeletal* OR exo OR end-effector OR haptic* OR robot regulation*	
**S2**	rehab* OR intervention* OR treatment* OR therap* OR program* OR strateg* OR training OR physiotherap* OR physio-therap* OR “physiotherap*” OR “physical therap*”	
**S3**	Qualitative research OR qualitative OR interview* OR focus group* OR ethno* OR phenomenolog* OR hermeneutic* OR grounded theory OR narrative analysis OR thematic analysis OR lived experience* OR life experience*	
**S4**	(“population group*” OR “patient* group*”) OR (men OR women OR patient* OR female OR male OR subjects)	
**S5**	“animal models” OR animal	
**Searched**	S1 AND S2 AND S3 AND S4 AND NOT S5	
**Applied Filters**		* Published*: 2021–2023
		*Language*: English

**Table 3 healthcare-13-00828-t003:** Our search strategy in Web of Science and applied filters.

**S1**	robotic* OR robot* OR robotic therap* OR robot-assisted OR robot assisted OR exoskeleton* OR assistive robotic* OR walking robotic device* OR personal care robot* OR medical robot* OR assistive OR assistive automation OR wearable robot* OR orthotic* OR orthosis OR exoskeletal* OR exo OR end-effector OR haptic* OR robot regulation*	
**S2**	rehab* OR intervention* OR treatment* OR therap* OR program* OR strateg* OR training OR physiotherap* OR physio-therap* OR “physiotherap*” OR “physical therap*”	
**S3**	Qualitative research OR qualitative OR interview* OR focus group* OR ethno* OR phenomenolog* OR hermeneutic* OR grounded theory OR narrative analysis OR thematic analysis OR lived experience* OR life experience*	
**S4**	(“population group*” OR “patient* group*”) OR (men OR women OR patient* OR female OR male OR subjects)	
**S5**	“animal models” OR animal	
**Searched**	S1 AND S2 AND S3 AND (S4 NOT S5)	
**Applied Filters**		* Published*: 2021–2023
		*Language*: English

## Data Availability

The data that support the findings are available from the first and corresponding author on reasonable request.

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
