# Peer review of "Towards the Identification of Patients’ Needs for Promoting Robotics and Allied Digital Technologies in Rehabilitation: A Systematic Review"

_healthcare, 2025, doi:10.3390/healthcare13070828_

Round 1
Reviewer 1 Report
Comments and Suggestions for Authors
Reviewer Comments:
- Introduction:
- The authors comment that there is a 2020 review on the same topic. They should explain the rationale for a new review in more detail. In this regard, they should build on the most relevant findings and conclusions of the previous review. As well as the future directions together with the limitations established by the authors of the previous review.
- Methods
- Why weren't more databases used to broaden the search?
- Results:
- Indicate reasons for exclusion of items in the flow chart
- A section on the main variables used or analysed by the studies could be included.
- The authors discuss the results, but could establish further differences with respect to the previous review. In addition, other reviews related to the topic in question could be included.
Author Response
1. Introduction:
Comment 1.1: The authors comment that there is a 2020 review on the same topic. They should explain the rationale for a new review in more detail. In this regard, they should build on the most relevant findings and conclusions of the previous review. As well as the future directions together with the limitations established by the authors of the previous review.
Response 1.1: We thank the Reviewer for highlighting this aspect. We have now revised our Introduction section, clarifying the rationale for conducting an updated review based on the findings of Laparidou et al. (2021). Given the rapid technological developments in rehabilitation robotics, there is the necessity of updating literature regularly. Our aim was therefore to extend the previous results to new literature.
Specifically, the previous systematic review synthesized qualitative studies exploring end-user experiences (patients, carers, and healthcare professionals) with robotic motor rehabilitation, identifying six core themes. Furthermore, Laparidou et al. explicitly recommended future research to systematically explore standardized methodologies and assessment tools for capturing user experiences more comprehensively, emphasizing the importance of structured frameworks to ensure consistency and comparability across studies. We investigated whether qualitative studies had structured their findings using the International Classification of Functioning, Disability, and Health (ICF) framework, as recommended for capturing a holistic view of disability, daily life activities, and participation.
Indeed, our systematic review was motivated by our goal to develop a structured survey for assessing patients’ needs and experiences within robotic rehabilitation and to use the acquired information for designing pragmatic clinical trials on robotic rehabilitation for different patients’ populations. The review of the recent literature concerning the methodologies of collecting end-users’ needs thus represents the initial step towards this goal.
In particular, given the heterogeneity of patient populations, clinical conditions, and rehabilitation goals involved in our future trials, our intention is to create a needs’ assessment tool based explicitly on the ICF. Utilizing the ICF framework would enable us to comprehensively capture dimensions related not only to motor impairments but also to patients' participation, perspectives, and daily-life functioning. Thus, reviewing recent literature on qualitative methodologies and the use of ICF-based tools was a foundational step in guiding the design and ensuring the validity and relevance of our patient-centred survey. As mentioned in the manuscript, we in fact observed a lack of studies employing ICF in literature.
In the revised version of the manuscript, we have explicitly discussed the most relevant conclusions drawn in Laparidou et al.’s review, including gaps in the inclusion of the structured frameworks and limitations related to the heterogeneity of studies and populations involved. We have also clarified that our review specifically aims to identify recent advancements in methodologies and tools for assessing patients’ and practitioners’ needs, as recommended by the future directions of Laparidou et al., by systematically exploring standardized methodologies and the explicit use of the ICF framework to ensure comprehensive assessment of patients' and practitioners' needs.
The Introduction section (pp.1-3, lines 33-126) now reads:
“The integration of technology in the rehabilitation field, such as end-effector robots, electromechanical devices, exoskeletons, wearable sensors, virtual and augmented reality, and others, has gained significant attention in recent years, as it offers the potential to enhance patient outcomes and promote personalized care [1–3]. However, the widespread adoption of robotics and other technology in rehabilitation has been limited so far. While results, especially from studies involving neurological patients, are encouraging, such technologies are still only accessible in a limited number of clinical centers, often as part of clinical trials.
To ensure effective integration of robotics into the clinical practice, it is vital to understand the diverse needs of patients who could potentially benefit from rehabilitation technology. Such needs can differ considerably in patients with sensory, motor, and/or cognitive deficits due to different diseases and impairments of varying degree. Matching these individualized needs with appropriate technology and intervention strategies is crucial to optimize clinical outcomes. A comprehensive understanding of patient needs is therefore fundamental for clinicians to determine the best strategies for implementing robotics and technological solutions effectively. Understanding the needs of healthcare practitioners is equally important, as their perceptions, expectations, and readiness to adopt new technologies significantly influence the successful implementation of these innovations in clinical settings. This knowledge not only informs the development of tailored interventions but also ensures that the technology is truly beneficial and aligns with both patients' and practitioners' expectations, thus facilitating broader acceptance, also among healthcare policy-makers, and successful integration into clinical practice.
The International Classification of Functioning, Disability, and Health (ICF) framework offers a standardized approach to assess health and functional status [4]. Since its adoption by the World Health Assembly in 2001, the ICF has been utilized across various settings for data collection and health assessments. It serves as a common language and data standard that enhances communication among healthcare providers, researchers, and policymakers [5]. By offering a structured, comprehensive, and versatile framework, the ICF significantly enhances the understanding and measurement of functioning and disability, ultimately improving rehabilitation research and practices. As part of this ongoing development, initiatives like the ICF Core Set Project have emerged [6], aiming to refine the application of the ICF across different health contexts and life stages, further solidifying its role as a vital tool in rehabilitation research and practice.
Moving beyond traditional medical perspectives, the ICF framework shifts attention from the causes of conditions to their impact on daily life, categorizing disability as impairments in body function or structure, activity limitations, and participation restrictions. In particular, it enables a comprehensive description of functional impairments across various domains (such as muscle strength, muscle tone, range of motion of joints, pain, balance), and different activities (such as mobility, writing, communicating, self-care), regardless of the underlying condition [7]. In fact, as highlighted in the literature [8], the acceptability, as well as recommendation for use, of rehabilitation devices often depends on the patient's level of disability in specific functional domains and activities. Using the ICF framework, rehabilitation needs can be explored integrating physical, social, and environmental aspects for a multidimensional assessment of care and interventions.
Rehabilitation tools can then be more accurately tailored to specific patient requirements, leading to better clinical outcomes. ICF-based documentation tools indeed support evidence-based practice by providing a structured approach to assessing patients' functioning, planning interventions, and evaluating outcomes [9,10]. ICF-based tools may also enable patients to express their rehabilitation priorities more effectively, leading to a greater sense of agency and self-determination [11]. Consequently, they would be better equipped to participate actively in their rehabilitation programs, which is crucial for sustained engagement and success.
Moreover, by offering a unified language and standardized framework, the ICF enhances communication among multidisciplinary teams. This standardization ensures consistency in assessing and documenting patient functioning across different healthcare settings and research studies [12,13].
A previous systematic review by Laparidou et al. (2020) [14] highlighted crucial insights from qualitative studies on end-user experiences with robotic rehabilitation, identifying six main analytical themes: logistic barriers, technological challenges, appeal and engagement, supportive interactions and relationships, benefits for physical, psychological, and social functioning, and expanding and sustaining therapeutic options. Despite the thorough thematic synthesis provided, the authors acknowledged important limitations in literature, particularly methodological heterogeneity across included studies and limited adoption of standardized frameworks for assessing patient experiences and needs. Laparidou et al. explicitly recommended future research to systematically explore standardized methodologies and assessment tools for capturing user experiences more comprehensively, emphasizing the importance of structured frameworks to ensure consistency and comparability across studies. As Laparidou et al. concluded, participants in the reviewed studies also made recommendations for future use and development of robotic devices and interventions. Systematically identifying and addressing patient and practitioner needs should directly guide the development of tailored, patient-centered rehabilitation technologies, as mentioned above.
The objective of this systematic review was therefore to provide a comprehensive investigation of the current literature on the methodologies used to collect data regarding end-users’ needs in robot- and technology-assisted rehabilitation. Specifically, we aimed to evaluate the different approaches utilized in these studies to understand patients' needs, and to identify gaps where further development or standardization is needed. To achieve this, we updated the systematic review by Laparidou et al. by extending its scope to include recent findings published between 2021 and 2023, in order to remain aligned with ongoing advancements in the robotic rehabilitation field. For all the above-mentioned reasons, particular emphasis was put here on tools based on the ICF, in order to comprehensively capture dimensions related not only to functional impairments (e.g., motor deficits) but also to patients’ participation, perspectives, and daily-life implications of their condition.
This work is part of the Italian Initiative “Fit for Medical Robotics” (Fit4MedRob) aimed at implementing robotics and allied digital technologies in clinical practice across different patients’ age groups, from childhood to elderly, affected by different diseases. As part of this Initiative, we plan to conduct pragmatic and exploratory trials to evaluate the effectiveness and sustainability of robotics-assisted rehabilitation. In this view, the systematic review presented here serves as a critical step towards the development of surveys designed to capture the needs of patients. These surveys will utilize the insights gathered in this review to more accurately assess and address the rehabilitation requirements of diverse patients’ populations, a foundational starting point to finally guide the design of relevant clinical trials on robotic rehabilitation within the scope of the Initiative.”
2. Methods
Comment 2.1: Why weren't more databases used to broaden the search?
Response 2.1:
We appreciate the Reviewer’s suggestion and acknowledge the importance of conducting a comprehensive literature search. Following this recommendation, we have expanded our search strategy to include Scopus and Web of Science in addition to PubMed. This extension has allowed us to broaden the scope of our review, ensuring a more exhaustive identification of relevant studies.
Consequently, we have updated the Methods and Results sections of the manuscript to reflect this revised search strategy, including details on the additional records retrieved and how they impacted the final selection of studies. We believe this modification enhances the robustness and completeness of our systematic review.
3. Results
Comment 3.1: Indicate reasons for exclusion of items in the flow chart
Response 3.1: We thank the Reviewer for the suggestion. We have taken steps to further clarify the study selection process. While our manuscript already included a statement in the Methods section specifying that the papers were excluded based on the inclusion criteria, we now explicitly indicate the reasons for exclusion both in the main text (p.7, lines 233-238) and in the flowchart (Fig.1, p.8).
Comment 3.2: A section on the main variables used or analysed by the studies could be included.
Response 3.2: We appreciate the Reviewer's insightful suggestion to include a section summarizing the main variables analyzed in the reviewed studies. In response, we have enhanced the Results section (pp.9-10, lines 309-323) with a dedicated paragraph that outlines the critical dimensions assessed through interviews and questionnaires in the selected studies:
“[…] the primary variables explored in the selected research studies for the period 2021-2023 can be categorized into:
- usability, safety, accessibility, user acceptance of the technology [40–42,45,47,49–52,54,57];
- user perceptions of effectiveness, personal experiences and satisfaction while using the technology [39–42,44–46,53–57];
- need for professional supervision [50];
- user requirements for the design of the technology [48,49,51];
- adaptability of the technology to home and community settings [42,49,50,52,53,57];
- social and cultural impacts of the technology, i.e., impacts on function, independence and dignity of users [43,46].”
Comment 3.3: The authors discuss the results, but could establish further differences with respect to the previous review. In addition, other reviews related to the topic in question could be included.
Response 3.3: We appreciate the Reviewer’s suggestion to include additional reviews related to the topic. However, after an extensive literature search, we found that there are no other systematic reviews specifically investigating the methodologies used to assess patients' and practitioners' needs in robotic rehabilitation or the application of structured frameworks such as the ICF in this context. Laparidou et al. (2021) identified key experiential themes in the literature related to patients’ and practitioners’ needs and expectations. Building upon their work, we have updated the literature search, while also taking a methodological perspective, focusing on how these needs are captured through qualitative tools rather than on the experiential themes. In particular, we sought to assess the extent to which standardized tools, such as the ICF framework, are integrated into current research.
We have added the following text to the Introduction section (p.3, lines 89-116):
“A previous systematic review by Laparidou et al. (2020) [14] highlighted crucial insights from qualitative studies on end-user experiences with robotic rehabilitation, identifying six main analytical themes: logistic barriers, technological challenges, appeal and engagement, supportive interactions and relationships, benefits for physical, psychological, and social functioning, and expanding and sustaining therapeutic options. Despite the thorough thematic synthesis provided, the authors acknowledged important limitations in literature, particularly methodological heterogeneity across included studies and limited adoption of standardized frameworks for assessing patient experiences and needs. Laparidou et al. explicitly recommended future research to systematically explore standardized methodologies and assessment tools for capturing user experiences more comprehensively, emphasizing the importance of structured frameworks to ensure consistency and comparability across studies. As Laparidou et al. concluded, participants in the reviewed studies also made recommendations for future use and development of robotic devices and interventions. Systematically identifying and addressing patient and practitioner needs should directly guide the development of tailored, patient-centered rehabilitation technologies, as mentioned above.
The objective of this systematic review was therefore to provide a comprehensive investigation of the current literature on the methodologies used to collect data regarding end-users’ needs in robot- and technology-assisted rehabilitation. Specifically, we aimed to evaluate the different approaches utilized in these studies to understand patients' needs, and to identify gaps where further development or standardization is needed. To achieve this, we updated the systematic review by Laparidou et al. by extending its scope to include recent findings published between 2021 and 2023, in order to remain aligned with ongoing advancements in the robotic rehabilitation field. For all the above-mentioned reasons, particular emphasis was put here on tools based on the ICF, in order to comprehensively capture dimensions related not only to functional impairments (e.g., motor deficits) but also to patients’ participation, perspectives, and daily-life implications of their condition.”
And the following text to the Discussion section (p.20, lines 340-345):
“This systematic review identified 19 new studies from 2021 to 2023, in addition to the 20 previously reviewed by Laparidou et al. (2020) [14], thus providing an updated perspective on current trends and limitations in this research area. Building upon the previous review, we also investigated whether qualitative studies had structured their findings using the ICF framework, as recommended for capturing a holistic view of disability, daily life activities, and participation.”
Reviewer 2 Report
Comments and Suggestions for Authors
This study aimed at the systematic review which used ICF way to identify the patients' needs for the robotic in rehabilitation. Here are comments for the authors:
- For a review paper, more literatures are necessary. Besdies, no more ICF-based researches in the new survey may result in the unclear purpose of this study. The advantages of ICF-based way shall be disclosed by the evidences which revealed in others' researches.
- This study is based on the previous review article(Laparidou et al. (2020)). The development difference of ICF-based researches between both is suggested to be disclosed in the paper.
- Conclusion shall response to the goals in Line 71-73. Please clear and simplify the study contribution. The first paragraph is not necessary.
Author Response
Comment 1 & 2:
For a review paper, more literatures are necessary. Besdies, no more ICF-based researches in the new survey may result in the unclear purpose of this study. The advantages of ICF-based way shall be disclosed by the evidences which revealed in others' researches.
This study is based on the previous review article(Laparidou et al. (2020)). The development difference of ICF-based researches between both is suggested to be disclosed in the paper.
Response 1 & 2:
We sincerely thank the Reviewer for the insightful feedback. We acknowledge that the connection between our review and the previous one by Laparidou et al. (2020), as well as the rationale for focusing on ICF-based research, may not have been sufficiently clear in our initial manuscript.
Our study was motivated by the need to develop a survey to assess patients' and practitioners' needs in robotic rehabilitation, particularly for a heterogeneous population of neurological patients. Given the diversity of conditions and rehabilitation goals, we sought a common framework capable of capturing not only motor impairments but also the broader impact of disability on daily life activities and social participation. The ICF was chosen because it provides a structured and multidimensional perspective that extends beyond a deficit-based approach to disability.
To inform the development of this survey, we conducted a literature review to evaluate whether and how recent studies have incorporated any structured framework for collecting patients’ needs (and in particular we looked for ICF-based tools), and to identify methodological advancements in qualitative research.
To better reflect this rationale, we have revised the Introduction section, explicitly clarifying the connection between our review and Laparidou et al. (2020), and we have included further references that highlight the advantages of using the ICF framework in rehabilitation research.
The Introduction section (pp.2-3, lines 55-116) now reads:
“The International Classification of Functioning, Disability, and Health (ICF) framework offers a standardized approach to assess health and functional status [4]. Since its adoption by the World Health Assembly in 2001, the ICF has been utilized across various settings for data collection and health assessments. It serves as a common language and data standard that enhances communication among healthcare providers, researchers, and policymakers [5]. By offering a structured, comprehensive, and versatile framework, the ICF significantly enhances the understanding and measurement of functioning and disability, ultimately improving rehabilitation research and practices. As part of this ongoing development, initiatives like the ICF Core Set Project have emerged [6], aiming to refine the application of the ICF across different health contexts and life stages, further solidifying its role as a vital tool in rehabilitation research and practice.
Moving beyond traditional medical perspectives, the ICF framework shifts attention from the causes of conditions to their impact on daily life, categorizing disability as impairments in body function or structure, activity limitations, and participation restrictions. In particular, it enables a comprehensive description of functional impairments across various domains (such as muscle strength, muscle tone, range of motion of joints, pain, balance), and different activities (such as mobility, writing, communicating, self-care), regardless of the underlying condition [7]. In fact, as highlighted in the literature [8], the acceptability, as well as recommendation for use, of rehabilitation devices often depends on the patient's level of disability in specific functional domains and activities. Using the ICF framework, rehabilitation needs can be explored integrating physical, social, and environmental aspects for a multidimensional assessment of care and interventions.
Rehabilitation tools can then be more accurately tailored to specific patient requirements, leading to better clinical outcomes. ICF-based documentation tools indeed support evidence-based practice by providing a structured approach to assessing patients' functioning, planning interventions, and evaluating outcomes [9,10]. ICF-based tools may also enable patients to express their rehabilitation priorities more effectively, leading to a greater sense of agency and self-determination [11]. Consequently, they would be better equipped to participate actively in their rehabilitation programs, which is crucial for sustained engagement and success.
Moreover, by offering a unified language and standardized framework, the ICF enhances communication among multidisciplinary teams. This standardization ensures consistency in assessing and documenting patient functioning across different healthcare settings and research studies [12,13].
A previous systematic review by Laparidou et al. (2020) [14] highlighted crucial insights from qualitative studies on end-user experiences with robotic rehabilitation, identifying six main analytical themes: logistic barriers, technological challenges, appeal and engagement, supportive interactions and relationships, benefits for physical, psychological, and social functioning, and expanding and sustaining therapeutic options. Despite the thorough thematic synthesis provided, the authors acknowledged important limitations in literature, particularly methodological heterogeneity across included studies and limited adoption of standardized frameworks for assessing patient experiences and needs. Laparidou et al. explicitly recommended future research to systematically explore standardized methodologies and assessment tools for capturing user experiences more comprehensively, emphasizing the importance of structured frameworks to ensure consistency and comparability across studies. As Laparidou et al. concluded, participants in the reviewed studies also made recommendations for future use and development of robotic devices and interventions. Systematically identifying and addressing patient and practitioner needs should directly guide the development of tailored, patient-centered rehabilitation technologies, as mentioned above.
The objective of this systematic review was therefore to provide a comprehensive investigation of the current literature on the methodologies used to collect data regarding end-users’ needs in robot- and technology-assisted rehabilitation. Specifically, we aimed to evaluate the different approaches utilized in these studies to understand patients' needs, and to identify gaps where further development or standardization is needed. To achieve this, we updated the systematic review by Laparidou et al. by extending its scope to include recent findings published between 2021 and 2023, in order to remain aligned with ongoing advancements in the robotic rehabilitation field. For all the above-mentioned reasons, particular emphasis was put here on tools based on the ICF, in order to comprehensively capture dimensions related not only to functional impairments (e.g., motor deficits) but also to patients’ participation, perspectives, and daily-life implications of their condition. ”.
Furthermore, we have modified our Results and Discussion sections to highlight that our review specifically investigated the aspect of ICF-based methodologies, and found that they remain largely absent in robotic rehabilitation research. The Results section (p.9, lines 305-311) now reads:
“Three studies only [20,53,55] applied the ICF framework to analyze the outcomes of their semi-structured interviews. In particular, Sivan et al. [20] structured the patient feedback collection process using relevant ICF categories; Forbrigger et al. [53] utilized the ICF to analyze the interaction between patient impairments and environmental factors for home-based device design; and Spits et al. [55] adopted the ICF to comprehensively explore patient experiences across body functions, activities, participation, and contextual domains.”
We have modified the Discussion section (p.20, lines 347-380) as follows:
“This systematic review identified 19 new studies from 2021 to 2023, in addition to the 20 previously reviewed by Laparidou et al. (2020) [14], thus providing an updated perspective on current trends and limitations in this research area. Building upon the previous review, we also investigated whether qualitative studies had structured their findings using the ICF framework, as recommended for capturing a holistic view of disability, daily life activities, and participation.
As shown, the literature employs mainly semi-structured interviews targeting small populations often with different pathologies and referring to the use of a single device. In particular, no studies used questionnaires/tools based on the ICF. Three studies only [20,53,55] applied ICF to analyze the outcomes of their assessments, and in [20] Authors asserted that the categories of the ICF Comprehensive Core Set could serve as a foundation for structuring interviews to capture user feedback. Nonetheless, as far as we are aware, they have not made such a tool accessible. The possibility of investigating through ICF the degree of impairment of different functions in a subject is an extremely important aspect, but the use of the ICF in robotic rehabilitation remains unexplored. Indeed, as emerged from the literature, the acceptability and usability of devices are strongly influenced by the severity and type of disability [43]. Given that functional impairments vary significantly across different domains, rehabilitation technologies should be tailored accordingly. The ICF provides a structured method to assess these aspects: the level of disability can be referred to individual functional domains—such as gait, balance, upper limb function, cognitive status, and speech— for patients with different diseases. A multidomain assessment would allow for a more individualized and precise adaptation of rehabilitation technologies. For instance, stroke patients with predominant gait impairments may require different technological interventions compared to those with cognitive deficits. Furthermore, stroke patients rarely experience impairments confined to a single domain, but they often present with a combination of motor, sensory, and cognitive deficits that affect their mobility, balance, coordination, executive functioning, and even language abilities. The ICF framework allows for classification across diverse health conditions while considering all these factors, as well as environmental and personal influences, which are crucial for designing patient-specific interventions. In fact, rehabilitation should not only be designed to restore movement but also to enhance functional independence and social participation. By failing to incorporate this framework, current research may lack the depth and consistency necessary for developing effective rehabilitation technologies that address, and are tailored to, the full spectrum of patient needs.”
Comment 3: Conclusion shall response to the goals in Line 71-73. Please clear and simplify the study contribution. The first paragraph is not necessary.
Response 3:
We thank the Reviewer for the suggestion. We have modified the Conclusion section (p.21, lines 424-432) also in consideration of another Reviewer’s suggestions, as follows:
“This systematic review examined the methodologies used to assess users’ needs in robotic rehabilitation and identified key gaps in standardization, such as the lack of structured assessment tools. While robotic technologies offer significant potential, the absence of systematic, validated methodologies limits our ability to comprehensively capture patients' and therapists' perspectives. To address these gaps, we propose developing tailored surveys grounded in the ICF framework, designed separately for patients and therapists to ensure that their distinct perspectives are accurately captured. These surveys will inform future research, enabling clinical trials better aligned to patients’ real-world needs, and eventually fostering the integration of robotics into clinical practice.”
Reviewer 3 Report
Comments and Suggestions for Authors
1. Line 33: Integration of which technologies?
2. Line 35: "...and other technology..." Which other technologies?
3. Lines 68-70: The aim of the paper is not related to the title?
4. Line 73-74: Since the literature review was done up to 2023, and today is February 2025, I believe that this research is outdated and should be expanded up to December 31, 2024.
5. The previous research (Laparidou et al. [6]) on which this paper is based was conducted in as many as 10 index databases. On the other hand, in this paper the authors consider only one identical database (MEDLINE), which is not comparable! Therefore, I suggest that the authors include at least two more databases: SCOPUS and Web of Science.
6. Line 118 and Table 1 (Applied Filters): “Only peer-reviewed studies in English, involving humans, were included.” Were only articles considered, without conf. papers, chapters, books, etc.? In order to verify the findings, all data must be explicitly stated, including the search method for each database individually (MEDLINE, SCOPUS and Web of Science), such as Title and/or Abstract, Full text, etc...
7. Figure 1: update in accordance with new databases; in each exclusion step, provide data for each database individually so that it is possible to determine the findings.
8. Conclusion: Conclude on the needs of patients and therapists, taking into account the considered robotic technologies in rehabilitation, their effectiveness and perspectives.
9. There are grammatical errors in several places, so I suggest the authors do a language review.
See the details in comments to authors.
Author Response
Comment 1. Line 33: Integration of which technologies?
Response 1. We thank the Reviewer for the comment. In the rehabilitation field, we refer to technologies such as end-effector robots, electromechanical devices, exoskeletons, virtual and augmented reality, wearable sensors, and AI-driven systems. We have clarified this in the revised version of the manuscript (p.1, lines 33-36):
“The integration of technology in the rehabilitation field, such as end-effector robots, electromechanical devices, exoskeletons, wearable sensors, virtual and augmented reality, and others, has gained significant attention in recent years, as it offers the potential to enhance patient outcomes and promote personalized care [1–3].”.
Comment 2. Line 35: "...and other technology..." Which other technologies?
Response 2. We apologize if this aspect was not clear in our manuscript. As mentioned above, in addition to robotics, we refer to technologies such as wearable sensors, exoskeletons, virtual and augmented reality, etc. We have specified this information in the revised version.
Comment 3. Lines 68-70: The aim of the paper is not related to the title?
Response 3. We thank the Reviewer for the comment. To address this, we have refined the objective statement as follows (p.3, lines 105-109):
“The objective of this systematic review was therefore to provide a comprehensive investigation of the current literature on the methodologies used to collect data regarding end-users’ needs in robot- and technology-assisted rehabilitation. Specifically, we aimed to evaluate the different approaches utilized in these studies to understand patients' needs, and to identify gaps where further development or standardization is needed.”
Comment 4. Line 73-74: Since the literature review was done up to 2023, and today is February 2025, I believe that this research is outdated and should be expanded up to December 31, 2024.
Response 4. We appreciate the Reviewer’s concern regarding the timeline of our literature review. However, given the nature of systematic reviews, updating the search at this stage would require a new comprehensive search and data synthesis, which we reckon not feasible at this stage.
That said, we recognize the importance of keeping research up to date. We already plan to conduct an updated review of studies up to 2025, ensuring a systematic two-year update from 2023. This will allow us to capture recent advancements and emerging methodologies in a structured and comprehensive manner.
Nonetheless, we have followed the suggestion of expanding the search to other databases.
Comment 5. The previous research (Laparidou et al. [6]) on which this paper is based was conducted in as many as 10 index databases. On the other hand, in this paper the authors consider only one identical database (MEDLINE), which is not comparable! Therefore, I suggest that the authors include at least two more databases: SCOPUS and Web of Science.
Response 5. We thank the Reviewer for the precious suggestion. We have conducted a new literature search in the SCOPUS and Web of Science databases and the results are presented in the revised manuscript.
Comment 6. Line 118 and Table 1 (Applied Filters): “Only peer-reviewed studies in English, involving humans, were included.” Were only articles considered, without conf. papers, chapters, books, etc.? In order to verify the findings, all data must be explicitly stated, including the search method for each database individually (MEDLINE, SCOPUS and Web of Science), such as Title and/or Abstract, Full text, etc...
Response 6. We thank the Reviewer for the comment. Only peer-reviewed articles in English involving humans were considered, including conference papers and book chapters, excluding only non-peer-reviewed sources and works without accessible full-text. We have revised the manuscript to explicitly state the search method for each database (MEDLINE, SCOPUS, and Web of Science), including the specific search terms and the use of “All Fields” for the query.
Comment 7. Figure 1: update in accordance with new databases; in each exclusion step, provide data for each database individually so that it is possible to determine the findings.
Response 7. We thank the Reviewer for the comment. We have updated Figure 1, including the studies collected and screened from the other two databases (Scopus and Web of Science). We now also provide reasons for exclusion in both the revised main text (p.7, lines 226-229) and the Figure 1 (p.8).
Comment 8. Conclusion: Conclude on the needs of patients and therapists, taking into account the considered robotic technologies in rehabilitation, their effectiveness and perspectives.
Response 8. We appreciate the Reviewer’s comment and understand the importance of concluding with a broader perspective on patient and therapist needs in relation to robotic rehabilitation technologies, their effectiveness, and future perspectives. On the other hand, we would like to clarify that the primary aim of our systematic review was not to assess the effectiveness of robotic technologies themselves, nor a meta-analysis of the specific needs of users, but rather to evaluate the methodologies used in the literature to capture patients’ and practitioners’ needs.
Our findings highlight the absence of structured, standardized assessment tools, particularly those based on the ICF framework, which limits the ability to systematically understand and address patients' and therapists' needs. Therefore, our conclusions emphasize the necessity of developing more structured assessment methods, ensuring that future robotic rehabilitation research and clinical practice are truly aligned with user requirements.
To reflect this, we have revised our Conclusion section (p.21, lines 423-431) ensuring alignment with the study’s aim while incorporating all Reviewers’ requests:
“This systematic review examined the methodologies used to assess users’ needs in robotic rehabilitation and identified key gaps in standardization, such as the lack of structured assessment tools. While robotic technologies offer significant potential, the absence of systematic, validated methodologies limits our ability to comprehensively capture patients' and therapists' perspectives. To address these gaps, we propose developing tailored surveys grounded in the ICF framework, designed separately for patients and therapists to ensure that their distinct perspectives are accurately captured. These surveys will inform future research, enabling clinical trials better aligned to patients’ real-world needs, and eventually fostering the integration of robotics into clinical practice.”
Comment 9. There are grammatical errors in several places, so I suggest the authors do a language review.
Response 9. We thank the Reviewer for the suggestion and we apologize for that. We have conducted a thorough language review to correct any grammatical errors and improve the clarity of the manuscript.
Round 2
Reviewer 3 Report
Comments and Suggestions for Authors
Since this is about advanced technologies and the systematic literature review was conducted up to 2023, and the authors did not expand the search.
Author Response
Comment 1: Since this is about advanced technologies and the systematic literature review was conducted up to 2023, and the authors did not expand the search.
Response 1: We sincerely thank the Reviewer for the comments and the time spent so far reviewing our manuscript.
This systematic review was carefully designed as the initial step in a broader, structured research Initiative named "Fit For Medical Robotics" (Fit4MedRob, https://www.fit4medrob.it/) funded by the Italian Ministry of Research and aimed at implementing robotics and allied digital technologies in clinical practice across different patients’ age groups and with different neuromotor and/or cognitive impairments. The insights from this review were intended to directly inform a subsequent phase involving the administration of online surveys to those patients, surveys that we have developed to capture their rehabilitative needs. The outcomes of these surveys then represented the guide for the design of pragmatic clinical trials, which are the core part of the Fit4MedRob Initiative (Mission 1), with the ultimate goal of demonstrating effectiveness and sustainability of the robotics and allied digital technologies in rehabilitation (as also stated in the manuscript). Indeed, our PROSPERO registration of the review explicitly refers to the review period ending in 2023, precisely because it aligns with the timeline of the Fit4MedRob project, ensuring methodological coherence.
This review is therefore intended to be the first in a series of structured, dedicated publications within Fit4MedRob, each building systematically upon the preceding work. Expanding the manuscript at this stage to incorporate additional data from 2024 and 2025 would require screening approximately 7000 additional articles, significantly delaying this manuscript and consequently the publication roadmap of the Fit4MedRob Initiative.
Given these considerations, we kindly advance our request, asking whether it would still be possible for the manuscript to be considered for publication in its current form, if every other aspect is in line with your Journal standards.